# Estimating encoding models of cortical auditory processing using naturalistic stimuli and transfer learning

**Nicolas Farrugia, Victor Nepveu, Deycy Camila Aryas Villamil**
Electronics Department
IMT Atlantique-LabSTICC
Brest, France
nicolas.farrugia@imt-atlantique.fr, victor.nepveu@imt-atlantique.net

## Abstract

The purpose of an encoding model is to predict brain activity given a stimulus. In this contribution, we attempt at estimating a whole brain encoding model of auditory perception in a naturalistic stimulation setting. We analyze data from an open dataset, in which 16 subjects watched a short movie while their brain activity was being measured using functional MRI. We extracted feature vectors aligned with the timing of the audio from the movie, at different layers of a Deep Neural Network pretrained on the classification of auditory scenes. fMRI data was parcellated using hierarchical clustering on 500 parcels, and encoding models were estimated using a fully connected neural network with one hidden layer, trained to predict the signals for each parcel from the DNN features. Individual encoding models were successfully trained and predicted brain activity on unseen data, in parcels located in the superior temporal lobe, as well as dorsolateral prefrontal regions, which are usually considered as areas involved in auditory and language processing. Taken together, this contribution extends previous attempts on estimating encoding models, by showing the ability to model brain activity using a generic DNN (ie not specifically trained for this purpose) to extract auditory features, suggesting a degree of similarity between internal DNN representations and brain activity in naturalistic settings.

## 1 Introduction

One important motivation for incorporating machine learning in neuroscientific discovery is the establishment of predictive models, as opposed to models based on statistical inference [1]. While the latter are unable to generalize to a new dataset, the former aim at sucessful generalization. In particular, encoding models aim at predicting brain activity given a model of the stimulus presented to the subject. A successful model should enable generalization to unseen data, enabling a better understanding of the underlying brain functions. Furthermore, an accurate encoding model could potentially be used to enhance machine learning, by providing an auxiliary source of training data, as recent evidence suggest that actual brain activity can guide machine learning [2]. In this study, we tested whether a pretrained network could be used to estimate encoding models, in the case of naturalistic auditory perception.

33rd Conference on Neural Information Processing Systems (NeurIPS 2019), Vancouver, Canada.

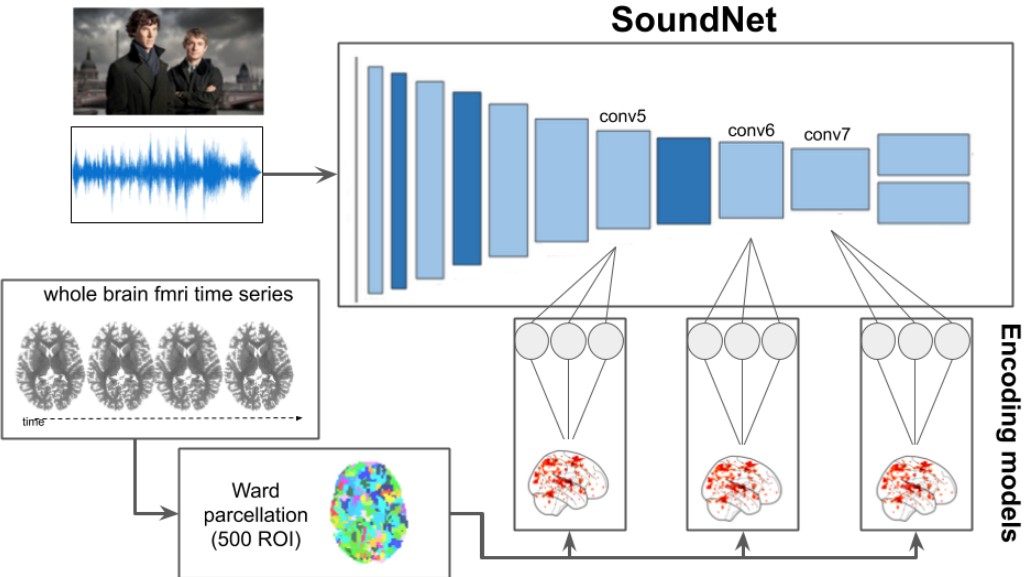

Figure 1: Overview of the proposed method.

## 2 Material and methods

### 2.1 Dataset

We downloaded the ds001110 dataset version 3 on openNeuro, in which 36 subjects watched a 20 minute long movie in an fMRI scanner [3]. MRI data was collected on a 3 T full-body scanner (Siemens Skyra) with a 20-channel head coil. Functional images were acquired using a $T2^*$-weighted echo planar imaging pulse sequence (TR 1500 ms, in-plane resolution 3 by 3 mm). Anatomical images were acquired using a T1-weighted magnetization-prepared rapid-acquisition gradient echo (MPRAGE) pulse sequence (0.89 mm3 resolution). More details can be found in the original paper [3]. In this study, we report only data from 16 subjects who watched the same episde from the TV show "Sherlock". At the time of submitting this paper, OpenNeuro included only raw data. Therefore, we used a preprocessed version[1].

### 2.2 Analysis

An overview of our method is presented in figure 1, and consists in three steps[2]. First, we extracted the audio track from the movie, and extracted feature vectors from all seven convolutional layers of SoundNet [6]. The 20 minute long audio file was fed as input, and we performed interpolations according to the width of the feature maps in each layer, in order to realign the obtained feature vectors with the temporal resolution (1.5 second) of the fMRI signal. This procedure yielded a total of 946 feature vectors, for each SoundNet layer.

Next, in order to reduce the dimensionality of the fMRI data, we applied hierarchical clustering using Ward criterion [4-5] to parcellate each individual brain into 500 regions of interests (ROI). We subsequently realigned the fMRI data on the beginning of the movie, yielding 946 vectors of dimension 500 for each subject.

Finally, encoding models were estimated separately for each subject and layer of SoundNet. We trained fully connected neural networks with one hidden layer to predict brain activity in the 500 ROI simultaneously. We report results when varying the number of neurons in the hidden layer. We used ReLu activation for the hidden layer, and linear activation for the output layer. We used a learning rate of 0.001 and a $L_2$ penalty of 0.0001. Cross validation on the data was performed using

---

[1]When we downloaded the data, openneuro had the preprocessed version available.

[2]Code at https://github.com/vnepveu/neurips19_neuroai_encoding

four folds without shuffling the data (in order to ensure that the train and test data were as distant as possible temporally), and for each training fold, $10\%$ of the data was kept for validation. We used the Adam optimiser with batches of size 50. Mean Square Error (MSE) was used as a loss function, and we applied an early stopping criterion that stops when validation MSE is not improving for 10 consecutive epochs. The final metric we use for evaluating the results is the $R^2$ score on the test set, indicating the quality of predicting fMRI data.

Additionally, we also perfomed a control analysis in which we estimated 100 null encoding models by extrating feature vectors from an *untrained* SoundNet, using the exact same procedure as described above. This analysis enables us to estimate the chance level of our dataset, as well as the gain obtained when using the pretrained network.

## 3   Results and discussion

### 3.1   Which layers enable the training of an encoding model ?

To begin with, as expected, the null models could not yield any significant training for any SoundNet layer, as indicated by $R^2$ scores of less than $1e^{-6} \pm 1e^{-5}$. Next, the first four layers of SoundNet could not be used to sucessfully train encoding models, as all $R^2$ were less than $0.03$. In the following analysis, we focus on results from layers conv5, conv6 and conv7.

Table 1 depicts the influence of the number of neurons in the hidden layer on the maximum over all ROI of $R^2$ scores , averaged across subjects and CV folds. The best results were obtained using 1000 neurons in the hidden layer, which enables succesfully training an encoding model on conv5, conv6 and conv7. Furthermore, we noticed that maximum $R^2$ scores across folds were much higher

| number of neurons | 50 | 100 | 500 | 1000 | 1500 |
|---|---|---|---|---|---|
| conv5 | 0.11 (0.08 ) | 0.13 (0.08) | 0.13 (0.08) | 0.13 (0.08) | 0.14 (0.08) |
| conv6 | 0.11 (0.07) | 0.11 (0.07) | 0.12 (0.07) | 0.12 (0.07) | 0.12 (0.07) |
| conv7 | 0.14 (0.15) | 0.18 (0.15) | 0.26 (0.13) | 0.28 (0.13) | 0.28 (0.12) |

Table 1: Hyper parameter exploration: average (standard error of the mean) across all subjects of maximum $R^2$ score, as a function of transfered layer, and number of neurons in hidden layer.

than the average over folds, and we found that the second fold consistently yielded low $R^2$ scores ($R^2 < 0.05$), for all subjects, for conv5 and conv6. Regarding conv7, the first two folds yielded an average of $R^2 = 0.15$ across all subjects. While this issue would demand closer inspection of the data, we suspect a systematic bias in the feature vectors for conv5 and conv6 in the first half of the video. Nonetheless, we obtained good generalization for the other three folds, especially for conv7. As a consequence, in the next section we will select the fold that yielded the maximum $R^2$ in order to interpret the spatial maps.

### 3.2   Where in the brain can feature vectors predict brain activity?

For all sixteen subjects, brain activity in ROIs including the superior and middle temporal gyri could be predicted with $R^2 > 0.25$. The corresponding SoundNet layer for which the $R^2$ was maximal was conv7 for 14 subjects, conv6 for one subject and conv5 for one subject. This results suggest that the information in the last layers of SoundNet is linked to the fMRI activity in regions previously associated with general purpose auditory processing.

Furthermore, for eleven out of sixteen subjects, brain activity in ROIs located in the left dorsolateral prefrontal cortex was predicted with $R^2 > 0.15$ (8 out of those eleven had a $R^2 > 0.25$). The corresponding layer was conv7 for 9 subjects and conv5 for 2 subjects. The left dorsolateral prefrontal cortex has been previously associated to verbal encoding in numerous studies, suggesting that the information in conv5 and conv7 might be linked to verbal content in the original stimuli, to a certain extent.

Note that we also found less than five subjects for which brain activity in medial regions of the Default Mode Network, such as the medial prefrontal cortex, or the anterior cingulate cortex, could be predicted, but this didn't seem like a consistent pattern across the majority of subjects. We depict

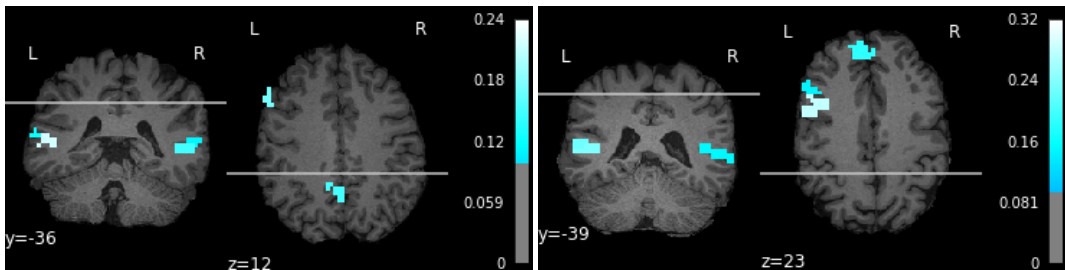

Figure 2: Threshold map of $R^2 > 0.1$ predicted by layer conv7 for two exemplar subjects, showing ROIs in the medial, superior temporal and left dorsolateral prefrontal cortices. We selected those exemplars to show the different possible cases.

in figure 2 exemplar subjects showing coronal and axial cuts of the $R^2$ maps, locating the ROIs in temporal, medial and dorsolateral prefrontal cortices.

### 3.3   General discussion, limitations and perspectives

We were able to train encoding models on individual subjects to predict brain activity using the deepest layers of SoundNet, using less than 20 minutes of fMRI data. The obtained models best predicted the activity in brain areas that are part of a language-related network. However, the current study has the following limitations. First, we extracted features from the auditory part of the stimuli, while the modeled brain activity involves many other brain functions, namely visual perception, as well as higher level cognitive functions such as memory and emotional responses. This probably explains why we obtain $R^2 = 0.5$ in the best case. Providing a richer stimuli representation using more general purpose feature extractors would probably enable a more complete model of brain activity. Second, we estimated brain parcellations on single subject data using only 20 minutes of MRI, which might not be enough to obtain a reliable set of ROIs [6]. Further studies should use either more repetitions on each subject, or attempt at learning parcellations across subjects, after having spatially normalized each individual to a template. Third, we didn't find a clear relationship between spatial extent of our encoding models as a function of the SoundNet layer. This could be due to the fact that SoundNet was trained independently of the brain data, and was never optimized for encoding models. One possible avenue would be to perform fine tuning, or retrain from scratch, in order to optimize the estimation of encoding models. Finally, in our approach we ignored the temporal dynamics of both the feature vectors and the fMRI data, as well as the dependencies between ROIs implied by brain connectivity. In future studies, we will consider the use of recurrent neural networks, as well as graph representation learning [7], in order to tackle those issues.

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
