# OpenReview forum: "Estimating encoding models of cortical auditory processing using naturalistic stimuli and transfer learning"
_NeurIPS.cc/2019/Workshop/Neuro_AI — Real Neurons & Hidden Units @ NeurIPS 2019 Poster_

### Official Review · AnonReviewer1 · 2019-09-19
**Straightforward application of DNN-based encoding models in the auditory domain**

**Clarity:** 3

**Comment:**

+ Use of audio-based DNNs can provide interesting insights into which stimulus properties drive neural responses
+ DLPFC results could potentially point to novel properties that are predictive of responses

- Novelty compared to existing work unclear
- Insights about what stimulus properties are driving the predicted responses will strengthen the paper

**Category:**

AI->Neuro

**Clarity Comment:**

Some typos. E.g. improvising => improving; this results; ...



**Evaluation:**

2: Poor

**Importance:**

2: Marginally important

**Importance Comment:**

This paper provides a straightforward application of SoundNet to predict fMRI responses to an audio stream. The question remains to what extent the presented results provide new results beyond those of https://www.cell.com/neuron/abstract/S0896-6273(18)30250-2 and https://arxiv.org/abs/1606.02627.

**Intersection:**

4: High

**Intersection Comment:**

Use of DNNs to explain observed brain responses falls right at the intersection.

**Rigor Comment:**

Standard statistics are performed. Thresholding for threshold map seems quite arbitrarily chosen. It remains unclear which stimulus features are driving the response predictions.

**Technical Rigor:**

2: Marginally convincing

---

### Official Review · AnonReviewer3 · 2019-09-24
**Deeper layers of a pre-trained auditory model can be used to partly predict fMRI responses, but not sure what we can conclude from that**

**Clarity:** 4

**Comment:**

I've given this the "good" evaluation because even though I personally am not convinced by this type of approach, it seems to be reasonably well done and I know that a number of people do find it useful and convincing.

**Category:**

AI->Neuro

**Clarity Comment:**

No problem understanding this work.

**Evaluation:**

3: Good

**Importance:**

2: Marginally important

**Importance Comment:**

Not sure how much we can gain from this sort of study. The authors show that deeper layers of a pre-trained auditory network can be used to predict fMRI responses, albeit not very well. But what can we conclude from this? Would the same be true of a different auditory network with very different properties? How much does it depend on the specific structure of that network? Could it just be the case that higher level features appear deeper in the network and correspond to areas recorded by fMRI?

**Intersection:**

4: High

**Intersection Comment:**

Definitely relevant, a similar approach to what has been tried with much success in vision.

**Rigor Comment:**

All seemed reasonable, but I would have liked to have seen controls against different architectures.

**Technical Rigor:**

3: Convincing

---

### Official Review · AnonReviewer2 · 2019-09-27
**Good start but some details unclear**

**Clarity:** 3

**Category:**

AI->Neuro

**Clarity Comment:**

The paper is understandable but some methods crucial methods details are left out (3.1 and 3.2)

**Evaluation:**

2: Poor

**Importance:**

3: Important

**Importance Comment:**

Understanding cortical acoustic processing is an important neuroscience goal. This paper doesn't however motivate the specific model being chosen, or what different layers mean. It is stated more like a prediction task instead of an understanding the brain task.

**Intersection:**

4: High

**Intersection Comment:**

Using an AI algorithm as a model of what the brain is doing. Although here how the model (SoundNet) could be an analogy of the brain (e.g. what could the different layers correspond to) could be more elaborated on.

**Rigor Comment:**

From what I understood, the R2 is being computed as the maximum over an ROI, at least in one part of the paper if not all. the maximum is a very noisy statistic and is not very reliable as a metric for model fitting and improvement.

From the paper, it seems that the authors picked the best fold to interpret (after looking at the results). This is effectively double dipping and negatively affects reproducibility.

R2 values in fMRI single trials are typically low and that is ok. The solution is not to use the maximum (if I understood correctly the motivation). The authors should be reporting single voxel metrics (over the brain) or should be computing some mean statistics.

**Technical Rigor:**

2: Marginally convincing

---

### Decision · Program_Chairs · 2019-10-02

Accept (Poster)